# Resource-Efficient Hybrid Quantum-Classical Simulation Algorithm

Chong Hian Chee,[1, *] Daniel Leykam,[1] Adrian M. Mak,[2] Kishor Bharti,[2, 3, †] and Dimitris G. Angelakis[1, 4, 5, ‡]

[1]*Centre for Quantum Technologies, National University of Singapore, 3 Science Drive 2, Singapore 117543*
[2]*A\*STAR Quantum Innovation Centre (Q.InC), Institute of High Performance Computing (IHPC), Agency for Science, Technology and Research (A\*STAR), 1 Fusionopolis Way, #16-16 Connexis, Singapore, 138632, Republic of Singapore.*
[3]*Centre for Quantum Engineering, Research and Education, TCG CREST, Sector V, Salt Lake, Kolkata 700091, India.*
[4]*School of Electrical and Computer Engineering, Technical University of Crete, Chania, Greece 73100*
[5]*AngelQ Quantum Computing, 531A Upper Cross Street, #04-95 Hong Lim Complex, Singapore 051531*
(Dated: October 14, 2024)

Digital quantum computers promise exponential speedups in performing quantum time-evolution, providing an opportunity to simulate quantum dynamics of complex systems in physics and chemistry. However, the task of extracting desired quantum properties at intermediate time steps remains a computational bottleneck due to wavefunction collapse and no-fast-forwarding theorem. Despite significant progress towards building a Fault-Tolerant Quantum Computer (FTQC), there is still a need for resource-efficient quantum simulators. Here, we propose a hybrid simulator that enables classical computers to leverage FTQC devices and quantum time propagators to overcome this bottleneck, so as to efficiently simulate the quantum dynamics of large systems initialized in an unknown superposition of a few system eigenstates. It features no optimization subroutines and avoids barren plateau issues, while consuming fewer quantum resources compared to standard methods when many time steps are required.

## I. INTRODUCTION

Quantum computers have long been touted as the natural solution to Hamiltonian simulation. They were first proposed by Feynman [1] to efficiently perform quantum time evolution and offer a provable exponential speedup over its classical counterparts [2]. Thus, Hamiltonian simulation is widely considered to have many practical applications in solving quantum dynamics of complex many-body systems in physics and chemistry [3, 4].

Many simulation algorithms developed over the past decades have relied on simulating the quantum time propagator $e^{-iHt/\hbar}$ directly on digital quantum computers [5–7]. Notable examples of digital simulations include the Trotter product decomposition [2, 8–13], linear combination of unitaries [14, 15], quantum walks [16, 17], quantum signal processing [18], qubitization [19], stochastic quantum simulation [20–22] and time-dependent quantum simulation in the interaction picture [23–26]. Whilst time propagation of the quantum states is an integral component to quantum dynamics, there is also the other important yet often unspoken task of extracting desired properties of the quantum states at intermediate times, as shown in Fig 1a. Computational bottlenecks in the standard method of simulating quantum dynamics include the preparation of multiple copies of the time-evolved quantum state at every time step, due to measurement "collapse of the wavefunction" [27] and the inability of state copies to be prepared quicker than their simulation time due to the no-fast-forwarding theorem [10, 28]. As a result, simulating long or rapidly oscillating dynamics become computationally expensive as more intermediate time step measurements are generally needed.

Quantum time propagation algorithms assume access to a Fault-Tolerant Quantum Computer (FTQC), which relies on a full implementation of quantum error correction [29–32]. This allows one to simulate any quantum system with arbitrary amounts of simulation time and error. In recent years, rapid development in quantum hardware has culminated in several intermediate-scale quantum error correction demonstrations using superconducting circuit [33], neutral-atom [34], trapped-ion [35] and silicon [36] platforms. Although these demonstrations have shown promising scaling capabilities to have more error-corrected qubits and gates, it still remains a challenging engineering endeavor to build a FTQC [37]. Thus, it is natural to seek algorithms from the Noisy Intermediate-Scale Quantum (NISQ) [38, 39] era for inspiration, owing to their frugal use of quantum resources that aims to minimize the effects of quantum noise.

Quantum-Assisted Simulator (QAS) [40, 41] is a hybrid quantum-classical simulation algorithm that was originally proposed to combine NISQ and classical computers to simulate quantum dynamics of a system with a time-independent Hamiltonian. It assumes a time-evolution ansatz as a linear combination of basis states. Using a classical computer, the linear coefficients are evolved via a set of complex linear dynamical equations derived from dynamical variational principles. Although QAS utilizes dynamical variational principles for simulation, it does not have any quantum variational optimization subroutines [39]. This allows the circumvention of the barren plateau problem [42, 43] as there is no need for any estimation of the variational landscape [40, 41]. Moreover, it guarantees energy conservation for simulations with time-independent Hamiltonians, that are distinct from

---

\* ch.chee@u.nus.edu
† kishor.bharti1@gmail.com
‡ dimitris.angelakis@gmail.com

most other variational quantum simulators in the literature [44–50]. The quantum computer is only used to calculate basis state overlaps, the Hamiltonian elements as defined in the dynamical equations and also the observable elements expressed in the basis states. The dynamical properties of the quantum state are then classically computed using the observable elements and the time-evolved linear coefficients. This avoids the computational bottleneck of preparing multiple copies of time-evolved quantum states for measurement at every intermediate time step.

In the original QAS proposal, however, the basis states were obtained by applying to the initial state different single Pauli-string operators generated from the Pauli Hamiltonian terms [40]. Despite being highly compatible with NISQ devices, such basis states are often linearly dependent and have basis set sizes that are unnecessarily larger than the entire quantum Hilbert space in order to ensure a good simulation fidelity [40, 51, 52]. QAS thus becomes inefficient for any quantum or classical computer to handle, as a result of exponential growth of the basis set to achieve desired fidelity.

In this work, we shall show how QAS can complement and utilize quantum time propagation algorithms implemented on FTQC devices to simulate quantum dynamics efficiently by considering time-evolved states themselves as a basis, as shown in Fig 1b. The time-evolved basis set size thus becomes equal to the time-evolution Hilbert subspace, where all time-evolved states reside. Specifically, if the system is initialized in an unknown superposition of $n$ system eigenstates, the QAS will only need to store and evolve $n$ linear coefficients, without needing any information on the eigenvalues and eigenstates. For a $2N$-qubit quantum device, QAS is expected to consume fewer quantum resources than the standard method when the number of time steps exceeds roughly $100n^2$. Therefore, if $n \ll 2^N$ is small, then QAS is able to perform more efficiently in simulating large $N$ system sizes, for a larger number of time steps than most existing hybrid quantum simulation algorithms. As a simple example, we shall demonstrate QAS for simulating quantum dynamics for a 4-qubit Helium atom and 8-qubit Hydrogen molecule initialized in a superposition of $n=2$ eigenstates.

## II. BACKGROUND

### A. Quantum-Assisted Simulator

We want to solve the quantum dynamics of a system, described by a time-independent Hamiltonian $\hat{H}$ and initialized in an unknown state $|\psi_0\rangle$, for an observable $\hat{O}$ at fixed times intervals $\Delta t$ up to a simulation time $T$. The quantum time evolution is described by the time-dependent Schrödinger equation,

$$i\hbar\partial_t |\Psi(t)\rangle = \hat{H} |\Psi(t)\rangle . \qquad (1)$$

QAS is a hybrid quantum-classical simulation al-

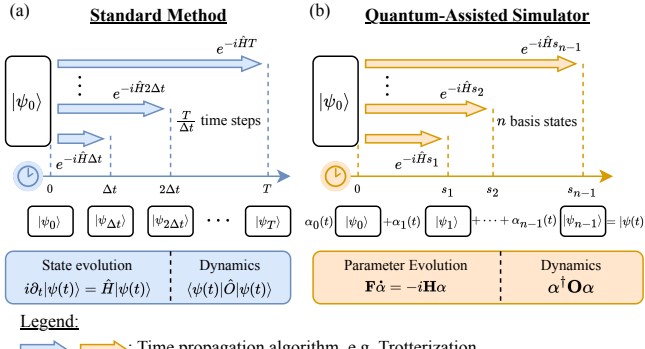

Figure 1. (a) Standard method to simulate the quantum dynamics of a system of Hamiltonian $\hat{H}$, initialized in $|\psi_0\rangle$, up to a time $T$, measured by an observable $\hat{O}$ at fixed time intervals $\Delta t$. (b) Our proposal to use Quantum-Assisted Simulator (QAS) to utilize quantum time propagation algorithms to efficiently simulate quantum dynamics. QAS assumes a time evolution state $|\psi(t)\rangle$ as a linear combination of time-evolved states. Atomic units, $\hbar=1$, are assumed in this figure.

gorithm that considers a time-evolution ansatz $|\psi(t)\rangle$ as a complex linear combination of $n_{\text{basis}}$ basis states $\{|\psi_j\rangle \,|\, j=0,1,\ldots,n_{\text{basis}}-1\}$ that is,

$$|\psi(t)\rangle = \sum_{j=0}^{n_{\text{basis}}-1} \alpha_j(t) |\psi_j\rangle , \qquad (2)$$

where the initial state $|\psi_0\rangle$ is part of the basis set, so that the initial linear coefficients can be simply written as $\boldsymbol{\alpha}(0)=(1,0,\ldots,0)$, thereby avoiding expensive initial state tomography. To ensure an accurate simulation, it is necessary for the basis set to be linearly independent and span the entire time-evolution Hilbert subspace – a subset of the full quantum Hilbert space where all possible time-evolved states reside. The choice of basis and the basis set size significantly determine the accuracy of the time-evolution.

After the basis set is chosen, QAS then splits the time-evolution task appropriately between a quantum and a classical computer. The quantum computer will compute the basis state overlaps, Hamiltonian, and observable elements in the chosen basis, whilst the classical computer stores and solves the dynamics of the complex linear coefficients in the ansatz in Eq. (2). The complex linear dynamical equation can be obtained by first applying McLachlan's dynamical variational principle [53] which states that the quantum time-evolution always minimizes the square root overlap between the left and right hand side states of the Schrödinger equation in Eq. (1). Then, by setting the variation of the square root overlap with respect to the linear coefficients $\boldsymbol{\alpha}$ to zero, that is,

$$\delta_{\boldsymbol{\alpha}} \| (i\hbar\partial_t - \hat{H}) |\psi(t)\rangle \| = 0, \qquad (3)$$

and substituting the ansatz from Eq. (2),

$$\boldsymbol{F}\dot{\boldsymbol{\alpha}} = -i\boldsymbol{H}\boldsymbol{\alpha}, \qquad (4)$$

where the elements of the matrices $\boldsymbol{F}$ and $\boldsymbol{H}$ are the basis state overlap and Hamiltonian matrix respectively,

$$F_{jk} = \langle \psi_j | \psi_k \rangle, \tag{5}$$

$$H_{jk} = \langle \psi_j | \hat{H} | \psi_k \rangle. \tag{6}$$

We refer readers to Appendix A for detailed derivation of parameter evolution in Eq. (4).

QAS can thus be summarized in four steps:

1. Choose a suitable basis set that includes the initial state for the simulation.

2. Use a quantum computer to estimate basis state overlap $\boldsymbol{F}$ in Eq. (5), Hamiltonian matrix $\boldsymbol{H}$ in Eq. (6), and the observable matrix $\boldsymbol{O}$, where $O_{jk} = \langle \psi_j | \hat{O} | \psi_k \rangle$. This can be done using a variety of quantum subroutines such as the Hadamard Test [54] or the projective measurements-based protocol [55].

3. Use a classical computer to solve the complex dynamical equations in Eq. (4) using a complex ordinary differential equation (ODE) solver to obtain the dynamics of the linear coefficients $\boldsymbol{\alpha}_t$.

4. Estimate the expectation value of observable $\langle \hat{O} \rangle$ at fixed time intervals $\Delta t$ up to a simulation time $T$ to solve the quantum dynamics,

$$\langle \psi(t) | \hat{O} | \psi(t) \rangle = \sum_{j,k=0}^{n_{\text{basis}}-1} \alpha_j^*(t) O_{jk} \alpha_k(t). \tag{7}$$

### B. Minimizing Quantum-Classical Resources

The hybrid quantum-classical setup in QAS enables certain types of simulations to run efficiently. In particular, consider a system of size $N$ initialized in the following unknown superposition of $n$ non-degenerate eigenstates of the system $|e_j\rangle$, where $n$ is defined as the number of eigenstates that span the time-evolution subspace of the quantum system, and assuming $n \ll 2^N$,

$$|\psi_0\rangle = \sum_{j=0}^{n-1} \beta_j |e_j\rangle. \tag{8}$$

Suppose we choose the following time-evolved states as a basis,

$$|\psi_j\rangle = e^{-i\hat{H}s_j} |\psi_0\rangle, \tag{9}$$

where $s_j$ are parameter times which are not more than the total simulation time, that is $0 = s_0 < s_1 < \ldots < s_{n_{\text{basis}}-1} \leq T$, using atomic units $\hbar=1$, which shall be used throughout this paper. Then, assuming linear independence of the basis, we will only need $n_{\text{basis}} = n$ time-evolved basis states that fully span the time-evolution Hilbert subspace. The quantum computer then handles these basis states with a dimension of $2^N$ each, whilst the classical computer will only need to store and evolve just $n_{\text{basis}} = n$ linear coefficients. If there are fewer time-evolved basis states than $n$, that is $n_{\text{basis}} < n$, the time-evolution ansatz is under-parameterized, leading to incorrect results. On the other hand, if the number of basis states is much larger than $n$, that is $n_{\text{basis}} > n$, the ansatz becomes over-parameterized which may lead to numerical convergence issues. The system eigenvalues, eigenstates $|e_j\rangle$, and the linear coefficients $\beta_j$ in the initial state in Eq. (8) remain unknown to us and need not be solved.

Assuming the system Hamiltonian $\hat{H}$ and observable $\hat{O}$ both decompose into a linear combination of $L$ number of Pauli strings $\hat{P}_l = \otimes_{j=1}^{N} \hat{\sigma}_j$ where $\hat{\sigma}_j \in \{\hat{I}_j, \hat{X}_j, \hat{Y}_j, \hat{Z}_j\}$, which may or may not share the same Pauli string, then getting the basis state overlap $\boldsymbol{F}$, Hamiltonian $\boldsymbol{H}$ and observable $\boldsymbol{O}$ matrices requires the estimation of the following quantities

$$F_{jk} = \langle \psi_0 | e^{i\hat{H}\Delta s_{jk}} | \psi_0 \rangle, \tag{10}$$

$$P_{jkl} = \langle \psi_0 | \hat{P}_l e^{i\hat{H}\Delta s_{jk}} | \psi_0 \rangle, \tag{11}$$

where $\Delta s_{jk} = s_j - s_k$ parameterizes the differences in the basis states' evolution times.

We first consider using the Hadamard Test with the time propagator $\hat{U} = e^{-i\hat{H}\Delta t}$ to evaluate a combined total of $\mathcal{O}(Ln^2)$ quantities, $F_{jk}$ and $P_{jkl}$ in Eqs. (10) and (11) respectively. An ancilla-controlled-$\hat{U}$ would have longer a quantum runtime complexity than the standard $\hat{U}$, but by no more than a constant multiple factor $\gamma$ [56]. If we now assume access to a $2N$-qubit quantum computer, we can consider a modified Hadamard test with $N$ ancilla qubits [56], instead of just one ancilla qubit considered previously. We refer readers to Appendix B for more details on the modified Hadamard Test. Controlled-quantum gates in the controlled-$\hat{U}$ that act on separate sets of systems qubits can remain parallel if the gates are controlled by separate ancilla qubits. In the worst case scenario, since all times $s_j$ are not more than the total simulation time $T$, all modified Hadamard tests have a maximum number of $\frac{T}{\Delta t}$ ancilla controlled-$U$s. The total overall quantum runtime complexity to estimate all the required quantities is thus $\mathcal{O}\left(\frac{T\gamma Ln^2}{\Delta t}\text{Poly}(L, \epsilon^{-1}, \Delta t)\right)$ where $\text{Poly}(L, \epsilon^{-1}, \Delta t)$ is the quantum runtime complexity of $\hat{U}$ and $\epsilon \geq \left\|e^{i\hat{H}\Delta t} - \hat{U}\right\|$ is the time evolution error.

In contrast, the standard method involves preparing $L$ copies of the time-evolved states and measuring each with the corresponding Pauli strings, decomposed from the observable, at every intermediate time step. Thus, this requires $\mathcal{O}\left(L\left(\frac{T}{\Delta t}\right)^2\right)$ repetitions of $\hat{U}$. The overall quantum runtime complexity to run the standard method is thus $\mathcal{O}\left(L\left(\frac{T}{\Delta t}\right)^2 \text{Poly}(L, \epsilon^{-1}, \Delta t)\right)$. We therefore estimate that QAS will consume fewer quantum re-

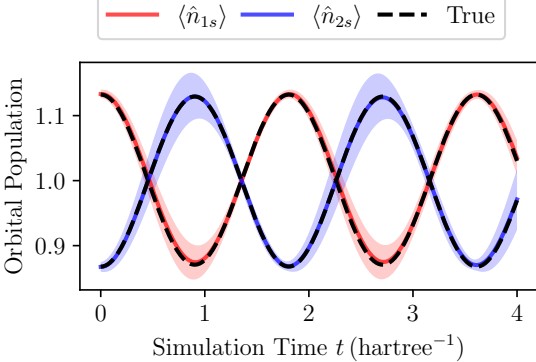

Figure 2. Atomic orbital population dynamics for the Helium atom using the 6-31G atomic orbital basis set, initialized in an equal superposition of the ground and highest excited state of eigenenergies -2.87 and 0.609 Hartrees respectively. The solid colored lines and shaded regions represents the mean and uncertainty of the 100 independent QAS simulation samples, each with $10^4$ simulated shots, respectively. The true time-evolution is denoted by the black dashed line.

sources than the standard method in terms of overall quantum runtime complexity, when the number of time steps exceeds $\frac{T}{\Delta t} \gtrsim 16\gamma n^2$. By making a reasonable assumption of $\gamma \approx 6$, based on the general observation that a 3-qubit Toffoli gate can be decomposed into at least 6 CNOT gates [57, 58], the threshold is approximated to be $\frac{T}{\Delta t} \gtrsim 100 n^2$. We refer readers to the full derivation of this claim in Appendix C.

After populating the necessary matrices, the complex linear dynamical equation in Eq. (4) is solved on a classical computer using a complex ODE solver with a $O\left(\frac{Tn^3}{\Delta t}\right)$ classical runtime complexity. Although QAS is computationally inefficient when $n \sim O(2^N)$, it becomes efficient when $n$ is small, that is $n \ll 2^N$, avoiding any exponential classical runtime or memory scaling. Hence, QAS is unsuitable for simulating the time-evolution of a system with arbitrary state initialization, it can be useful when simulating large systems initialized in a superposition of a low number of eigenstates $n$ and a large number of time steps that is more than $100n^2$.

### III. EXAMPLES AND RESULTS

We provide two examples in the context of quantum chemistry for a simple QAS demonstration: the orbital population dynamics of a Helium (He) atom and a hydrogen (H$_2$) molecule at an equilibrium bond distance of 1.4 Bohr, using the 6-31G atomic basis set, initialized to an equal superposition of ground and highest excited eigenstate. Both chemical systems can be described by the following second-quantized electronic Hamiltonian $\hat{H}_{\text{elec}}$ [59],

$$\hat{H}_{\text{elec}} = \sum_{pq}^{N} h_{pq} \hat{a}_p^\dagger \hat{a}_q + \frac{1}{2} \sum_{pqrs}^{N} h_{pqrs} \hat{a}_p^\dagger \hat{a}_q^\dagger \hat{a}_r \hat{a}_s, \quad (12)$$

where $\hat{a}_p^\dagger$ and $\hat{a}_p$ are fermionic creation and annihilation operators respectively for the $p^{\text{th}}$ atomic/molecular spin-orbital, $h_{pq}$ are one-electron core integrals, and $h_{pqrs}$ are two-electron repulsion integrals. The orbital population observable is the sum of spin-up and spin-down number operators $\hat{a}_\uparrow^\dagger \hat{a}_\uparrow + \hat{a}_\downarrow^\dagger \hat{a}_\downarrow$ that act on the corresponding orbital.

We implemented a numerical statevector calculation using the Jordan-Wigner (JW) fermion-to-qubit mapping [60], which transforms $\hat{H}_{\text{elec}}$ into a Pauli Hamiltonian $\hat{H}_P$ with up to $L \sim O(N^4)$ terms in general [4, 61]. $\hat{H}_{\text{elec}}$ for the He atom is mapped to a 4-qubit system with 27 terms in $\hat{H}_P$, and that for H$_2$ molecule is mapped to an 8-qubit system with 185 terms in $\hat{H}_P$. The time-evolved basis set consist of $n=2$ states: the initial state $|\psi_0\rangle$ and $|\psi_1\rangle = e^{-i\hat{H}/2} |\psi_0\rangle$, where we set the parameter time $s_1 = \frac{1}{2}$. Assuming an ideal noiseless Hadamard Test with $10^4$ simulated shots, we randomly sampled 100 sets of the basis state overlaps $\boldsymbol{F}$, Hamiltonian matrices $\boldsymbol{H}$ and the orbital population observable $\boldsymbol{O}$. Next, we solve the ODE according to Eq. (4) for every sample pair of $\boldsymbol{F}$ and $\boldsymbol{H}$ independently up to a simulation time $t=4$ Hartree$^{-1}$ with a time interval of $\Delta t=0.001$ Hartree$^{-1}$. The total number of time steps in this simulation is 4000, which far exceeds the threshold of $100 \cdot 2^2 = 400$ time steps, placing this QAS demonstration well into quantum resource-efficient regime. Finally, we calculated the orbital population at every time step for each sample run. We refer readers to Appendix D for details on how the sampling of $\boldsymbol{F}$, $\boldsymbol{H}$ and $\boldsymbol{O}$ matrices was performed.

We plotted the orbital population dynamics of the He atom and H$_2$ molecule in Figs. 2 and 3 respectively. The colored solid lines and shaded regions represents the population mean and uncertainty of the 100 independent QAS simulation samples, respectively. The true time-evolution is denoted by a black dashed line. In addition, we also refer readers to Appendix E for plots of the total, coulomb, kinetic and potential energy dynamics, which demonstrate the total energy conservation feature of QAS. For both systems, we observe the orbital population oscillates with a frequency of roughly 0.5 Hartree. This is in agreement with the true frequency of -0.554 and -0.490 Hartrees, respectively. For the helium atom, the maximum fractional population uncertainty is about 4% for the 1$s$ and 5% for the 2$s$ atomic orbitals. For the hydrogen molecule, the maximum fractional population uncertainty is about 6% for both 1$\sigma$ and 2$\sigma*$ molecular orbitals. We also observe that the population estimation of 1$\sigma*$ and 2$\sigma$ molecular orbitals are extremely imprecise due to its low orbital population. The low precision can be improved by simply increasing number of shots, for which we refer reader to Appendix F which details

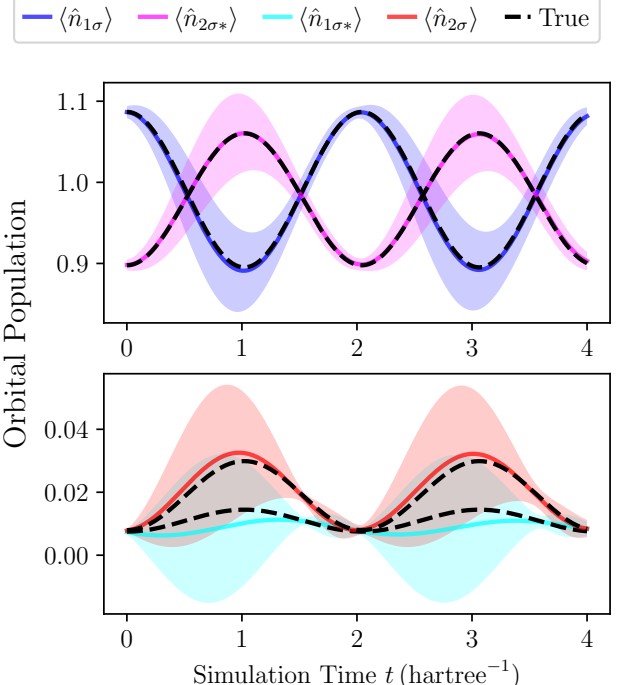

Figure 3. Molecular orbital population for Hydrogen molecule using the 6-31G atomic orbital basis set, at equilibrium distance of 1.4 Bohr, initialized in an equal superposition of the ground and highest excited state of eigenenergies -1.15 and 1.93 Hartrees, respectively. Solid colored line and shaded regions represent the mean and uncertainty of the 100 independent QAS simulation samples, each with $10^4$ simulated shots, respectively. The true time-evolution is denoted by the black dashed line.

the linear relationship between the variance of the estimated quantities and number of shots. In principle, if the quantum subroutines are executed with minimal and unbiased quantum noise, that is indistinguishable to statistical shot noise, then the QAS can accurately simulate the quantum dynamics, though more shots are needed to improve its precision.

## IV. DISCUSSION AND OUTLOOK

We have shown that by using a linear combination of time-evolved basis states as the evolution ansatz, QAS can simulate the quantum dynamics of large systems more efficiently than the standard method of preparing multiple copies of time-evolved states beyond a small number of time steps. More precisely, if the system is initialized in an unknown superposition of $n$ system eigenstates, then the QAS requires fewer quantum resources when the number of time steps exceeds $100n^2$, regardless of the system size $N$, the number of terms in the electronic Hamiltonian or the mapped Pauli Hamiltonian, and without any prior knowledge of the eigenstates and eigenvalues.

While we have demonstrated how a digital FTQC devices can be used by QAS to simulate quantum dynamics, an analog quantum device may be used as well due to its potential for practical quantum advantage in quantum simulation [62]. Several possible QAS approaches for analog quantum computation includes analog emulation of digital quantum simulation [63] or Hamiltonian learning techniques [64] to estimate elements of QAS dynamical equations in Eq. 4. However, such approaches must be further refined to mitigate any additional overheads that negates the efficiency of QAS.

Our results have demonstrated that useful dynamical quantities of chemical systems can be estimated fairly accurately and precisely with modest amount of quantum-classical resources and minimal heuristics. However, the QAS performance depends heavily on the quality of the quantum time propagation algorithm employed. As an example, we refer readers to Appendix G where we show how a large number of Trotter steps from the first-order Trotterization of time-evolved basis is required to achieve good simulation fidelity for the helium atom case. The trade-offs between the quality of the time-evolved basis state and the overall quantum runtime complexity must be thoroughly analyzed. Thus, an important direction would be to identify and refine time propagation techniques that would minimize quantum resources while maximizing simulation accuracy for different system types.

Whilst QAS may seem very accurate when compared with other simulation algorithms, there is a major caveat that needs to be pointed out. The time-evolved basis is often linearly dependent, making the complex dynamical equation in Eq. (4) prone to ill-conditioning, thus resulting in numerical instabilities. As the number of eigenstates in the initial state $n$ increases, there is an increasing likelihood of the time-evolved basis becoming too similar to each other as the corresponding eigenbasis amplitudes only differ in its phase angles but not in its magnitude, resulting in linear dependence of the basis. The parameter times must then be chosen carefully to avoid such an issue. In the $n=2$ base case, any parameter time $s_1 \neq \frac{2\pi k}{\Delta e}$, for any $k \in \mathbb{Z}$ and eigenvalue difference $\Delta e$, is acceptable as it leads to a well-conditioned problem, as shown in Appendix H. For the $n>2$ case, however, solving for the complete set of conditions for the parameter times becomes an incredibly difficult problem and remains an open question on the best method to choose a optimal set of parameter times. In practice, for small $n$, a suitable set of parameter times is chosen heuristically via trial and error to avoid the complexity of solving the conditions so as to ensure an accurate and efficient simulation.

The resource analysis has suggested that if $n \ll 2^N$ is kept significantly smaller than the size of the Hilbert space, then QAS becomes a practical hybrid quantum-classical algorithm as the classical runtime and memory scales polynomially in $n$. Quantum dynamical problems involving the dynamics of a few (unknown) eigenstates

that may particularly benefit from QAS implementation. For example, in systems hosting quantum many-body scars, special simple initial states termed scar states can be expressed as a superposition of a few eigenstates with dynamics restricted to a small subspace of the full Hilbert space [65, 66]. More generally, one may envisage quantum simulation applications involving systems whose ground state is resonantly excited by a fixed frequency drive term that is briefly applied, leading to dynamics involving eigenstates resonantly excited by the periodic drive term and its harmonics.

Besides applications to static systems, QAS can be extended to systems with a time-dependent Hamiltonian such as light-matter interaction, and atomic and molecular dynamics, though such tasks will likely involve a quantum-classical feedback loop which updates the Hamiltonian matrix $H$ at every simulation time step, which will significantly increase the computation runtime and quantum resources. Nevertheless, to realize such applications, an important direction would be to develop efficient quantum state preparation algorithms which can reliably generate interesting superpositions of a few eigenstates.

It is expected that digital quantum computers with dozens of error-corrected qubits and gates will be available for use in the near future. If so, the aforementioned quantum time propagation algorithms may potentially be demonstrated beyond trivial toy systems to small, yet interesting systems. However, there is still a need for quantum resource-efficient simulation algorithms even when FTQC devices are available. Hence, QAS may thereby achieve a "quantum-assisted advantage" in the simulation of quantum dynamics, realizing a practical end-to-end quantum solution for quantum simulation as first envisioned by Feynman.

## DATA AVAILABILITY STATEMENT

The data generated and/or analyzed during the current study are not publicly available for legal/ethical reasons but are available from the corresponding author on reasonable request.

## ACKNOWLEDGMENTS

This research is supported by the National Research Foundation, Singapore, and A*STAR under its CQT Bridging Grant and Quantum Engineering Programme, Grant No. NRF2021-QEP2-02-P02, A*STAR C230917003, by the European Union's Horizon Programme (HORIZON-CL4-2021-DIGITALEMERGING-02-10), Grant Agreement No. 101080085, QCFD.

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

## Appendix A: Derivation of Parameter Evolution

We present the derivation of the parameter evolution, adapted from Bharti and Haug [40], and Yuan et al. [44]. In the main text, we define the time-evolution state ansatz as

$$|\psi(t)\rangle = \sum_{j=0}^{n_{\text{basis}}-1} \alpha_j(t) |\psi_j\rangle. \tag{A1}$$

The McLachlan's variational principle [53] states that the quantum time-evolution always minimizes the square root error overlap between the left and right hand side states of the Schrödinger equation,

$$\delta_{\boldsymbol{\alpha}} \| (i\hbar\partial_t - \hat{H}) |\psi(t)\rangle \| = 0. \tag{A2}$$

where error overlap $\epsilon$,

$$\epsilon = \| (i\hbar\partial_t - \hat{H}) |\psi(t)\rangle \|^2 \tag{A3}$$

$$= [(i\hbar\partial_t - \hat{H}) |\psi(t)\rangle]^\dagger [(i\hbar\partial_t - \hat{H}) |\psi(t)\rangle] \tag{A4}$$

$$= \left\{ \sum_{j,k=0}^{n_{\text{basis}}-1} [\partial_{\alpha_j^*} \langle\psi(t)|][\partial_{\alpha_k} |\psi(t)\rangle] \dot{\alpha}_j^* \dot{\alpha}_k \right\} + i \left\{ \sum_{j=0}^{n_{\text{basis}}-1} [\partial_{\alpha_j^*} \langle\psi(t)|] \hat{H} |\psi(t)\rangle \dot{\alpha}_j^* \right\}$$

$$- i \left\{ \sum_{k=0}^{n_{\text{basis}}-1} \langle\psi(t)| \hat{H} [\partial_{\alpha_k} |\psi(t)\rangle] \dot{\alpha}_k \right\} + \langle\psi(t)| \hat{H}^2 |\psi(t)\rangle. \tag{A5}$$

Since,

$$\partial_{\alpha_j} |\psi(t)\rangle = |\psi_j\rangle, \tag{A6}$$

the error overlap $\epsilon$ in Eq. (A5) simplifies to

$$\epsilon = \left\{ \sum_{j,k=0}^{n_{\text{basis}}-1} \langle\psi_j|\psi_k\rangle \dot{\alpha}_j^* \dot{\alpha}_k \right\} + i \left\{ \sum_{j=0}^{n_{\text{basis}}-1} \langle\psi_j| \hat{H} |\psi(t)\rangle \dot{\alpha}_j^* \right\}$$

$$- i \left\{ \sum_{k=0}^{n_{\text{basis}}-1} \langle\psi(t)| \hat{H} |\psi_k\rangle \dot{\alpha}_k \right\} + \langle\psi(t)| \hat{H}^2 |\psi(t)\rangle \tag{A7}$$

$$= \dot{\boldsymbol{\alpha}}^\dagger \boldsymbol{F} \dot{\boldsymbol{\alpha}} - i\dot{\boldsymbol{\alpha}}^\dagger \boldsymbol{H} \boldsymbol{\alpha} + i\boldsymbol{\alpha}^\dagger \boldsymbol{H} \dot{\boldsymbol{\alpha}} + \boldsymbol{\alpha}^\dagger (\boldsymbol{H}^2) \boldsymbol{\alpha}, \tag{A8}$$

where have used the following matrices

$$F_{jk} = \langle\psi_j|\psi_k\rangle, \tag{A9}$$

$$H_{jk} = \langle\psi_j| \hat{H} |\psi_k\rangle, \tag{A10}$$

$$(\boldsymbol{H}^2)_{jk} = \langle\psi_j| \hat{H}^2 |\psi_k\rangle. \tag{A11}$$

As the variation of the square root error overlap is equivalent to the variation of the error overlap up to a constant factor, we may focus on variation of the error overlap instead, then we have

$$\delta_{\alpha_k} \epsilon = \left( \sum_{j=0}^{n_{\text{basis}}-1} [\partial_{\alpha_j^*} \langle\psi(t)|][\partial_{\alpha_k} |\psi(t)\rangle] \dot{\alpha}_j + i[\partial_{\alpha_k} |\psi(t)\rangle] \hat{H} |\psi(t)\rangle \right) \delta\dot{\alpha}_k^*$$

$$+ \left( \sum_{j=0}^{n_{\text{basis}}-1} [\partial_{\alpha_j^*} \langle\psi(t)|][\partial_{\alpha_k} |\psi(t)\rangle] \dot{\alpha}_j^* - i[\partial_{\alpha_k} |\psi(t)\rangle] \hat{H} |\psi(t)\rangle \right) \delta\dot{\alpha}_k \tag{A12}$$

$$= (\boldsymbol{F} \dot{\boldsymbol{\alpha}} + i\boldsymbol{H} \boldsymbol{\alpha}) \delta\dot{\alpha}_k^* + (\boldsymbol{F} \dot{\boldsymbol{\alpha}}^* - i\boldsymbol{H} \boldsymbol{\alpha}) \delta\dot{\alpha}_k. \tag{A13}$$

Thus, the parameter evolution is

$$\boldsymbol{F} \dot{\boldsymbol{\alpha}} = -i\boldsymbol{H} \boldsymbol{\alpha}, \tag{A14}$$

which minimizes the error overlap $\epsilon$ in Eq. (A8) throughout the entire evolution,

$$\epsilon_{\min} = \boldsymbol{\alpha}^\dagger (\boldsymbol{H}^2) \boldsymbol{\alpha} - \dot{\boldsymbol{\alpha}}^\dagger \boldsymbol{F} \dot{\boldsymbol{\alpha}}. \tag{A15}$$

## Appendix B: Modified Hadamard Test

The standard Hadamard Test estimates the real and imaginary components of $\langle\psi|\hat{U}|\psi\rangle$ using an ancilla qubit and an $N$-system-qubit unitary $\hat{U}$ gate controlled by the ancilla qubit. This creates a problem: parallel gates in the gate decomposition of $\hat{U}$ must be serialized when gate control by the ancilla qubit is applied. We shall show that this problem can be solved using more ancilla qubits. As a simple example, let $\hat{U}=\hat{U}_1\otimes\hat{U}_2$ where $\hat{U}_1$ and $\hat{U}_2$ act on two separate sets of system qubits. Then, we may consider the following modified Hadamard Test that uses two ancilla qubits, as shown in Fig. 4.

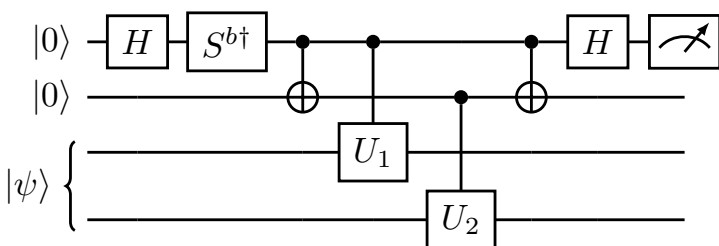

Figure 4. Modified Hadamard Test with two ancilla qubits initialized in $|0\rangle$ and two sets of system qubits initialized in $|\psi\rangle$.

The first three quantum gates on the left of the modified Hadamard test circuit in Fig. 4 generate an ancilla Bell state $|00\rangle+|11\rangle$ if $b=0$ or $|00\rangle-i|11\rangle$ if $b=1$, where we drop state normalization to reduce verbosity. Next, $\hat{U}_1$ controlled by the top ancilla qubit and $\hat{U}_2$ controlled by the bottom ancilla qubit are applied to prepare $|00\rangle|\psi\rangle+|11\rangle\hat{U}|\psi\rangle$ if $b=0$ or $|00\rangle|\psi\rangle-i|11\rangle\hat{U}|\psi\rangle$ if $b=1$. Then, the inverse of operator that generates the ancilla Bell state, without any phase gates $S$, is applied. Finally, the top ancilla qubit is measured in the Pauli-Z basis. The Pauli-Z expectation value $\langle Z\rangle$ will give the real and imaginary components of $\langle\psi|\hat{U}|\psi\rangle$ for $b=0$ and 1 respectively. To maximize gate parallelism, it is sufficient to have $N$ ancilla qubits. Doing so will require one to prepare a $N$-qubit ancilla GHZ state that is $|0\cdots0\rangle+|1\cdots1\rangle$ if $b=0$ or $|0\cdots0\rangle-i|1\cdots1\rangle$ if $b=1$, instead of an ancilla Bell state.

## Appendix C: Quantum Resource-Efficient Regime

Here we shall compare the total quantum runtime complexity between the standard method and the Quantum-Assisted Simulator (QAS) and derive a condition that QAS has to fulfill in order to be more quantum resource-efficient than standard methods. We are given the problem of solving the quantum dynamics of a system of qubit size $N$, described by a time-independent Hamiltonian $\hat{H}$, initialized in an unknown state $|\psi_0\rangle$, measured by an observable $\hat{O}$ at fixed time intervals $\Delta t$ up to a simulation time $T$.

We assume the system Hamiltonian $\hat{H}$ and observable $\hat{O}$ both decompose into a linear combination of $L$ Pauli strings $\hat{P}_l=\otimes_{j=1}^N\hat{\sigma}_j$, where $\hat{\sigma}_j\in\{\hat{I}_j,\hat{X}_j,\hat{Y}_j,\hat{Z}_j\}$, which may or may not share the same Pauli string. The quantum runtime complexity of existing quantum algorithms that simulates the $e^{-i\hat{H}\Delta t}$ time propagator up to a time evolution error $\epsilon\geq\left\|e^{i\hat{H}\Delta t}-\hat{U}\right\|$ are upper bounded by $\mathcal{O}(\text{Poly}_{\hat{U}}(L,\epsilon^{-1},\Delta t))$ as shown in Table I. Thus, performing the time evolution for a simulation time of $j\Delta t$, where $j\in\mathcal{Z}$ is a non-negative integer, incurs a quantum runtime complexity of $j\cdot\mathcal{O}(\text{Poly}_{\hat{U}}(L,\epsilon^{-1},\Delta t))$ from applying the $e^{-i\hat{H}\Delta t}$ time propagator $j$ times on the initial state.

Due to wavefunction collapse and no-fast-forwarding theorem, the standard method involves preparing $L$ copies of time-evolved states at times $\{j\Delta t|j=0,1,\ldots,\frac{T}{\Delta t}\}$ for Pauli measurements. Since, in the main text, we assume that QAS have access to a $2N$-qubit quantum computer, we may parallelize the the standard method by running two independent time-evolution at a time, reducing the overall runtime by a factor of 2. Therefore, the overall quantum runtime complexity of the standard method is

$$\frac{1}{2}\cdot L\cdot\sum_{j=0}^{T/\Delta t}\left[j\cdot\mathcal{O}(\text{Poly}_{\hat{U}}(L,\epsilon^{-1},\Delta t))\right]=\frac{1}{4}\frac{T}{\Delta t}\left(\frac{T}{\Delta t}+1\right)\cdot L\cdot\mathcal{O}(\text{Poly}_{\hat{U}}(L,\epsilon^{-1},\Delta t)) \tag{C1}$$

| Algorithms Simulating $e^{-i\hat{H}t}$ | Quantum runtime complexity |
|---|---|
| 1st Order Trotter [2, 8, 9] | $\mathcal{O}\left[L^3(t\|\hat{H}\|_{\max})^2/\epsilon\right]$ |
| 2nd Order Trotter [2, 8, 9] | $\mathcal{O}\left[L^{\frac{5}{2}}(t\|\hat{H}\|_{\max})^{\frac{3}{2}}/\epsilon^{\frac{1}{2}}\right]$ |
| 2kth Order Trotter [2, 10] | $\mathcal{O}\left[5^{2k}L(Lt\|\hat{H}\|_{\max})^{1+\frac{1}{2k}}/\epsilon^{\frac{1}{2k}}\right]$ |
| Qubitization [19] | $\mathcal{O}\left[t\lambda + \log(1/\epsilon)/\log\log(1/\epsilon)\right]$ |
| Linear Combination of Unitaries [15] | $\mathcal{O}\left[t\lambda\log(t\lambda/\epsilon)/\log\log(t\lambda/\epsilon)\right]$ |
| Quantum Signal Processing [18] | $\mathcal{O}\left[t\|\hat{H}\|_{\max} + \log(1/\epsilon)/\log\log(1/\epsilon)\right]$ |
| Stochastic Simulation (QDRIFT) [21] | $\mathcal{O}\left[(t\lambda)^2/\epsilon\right]$ |

Table I. The quantum runtime or gate complexity of various quantum algorithms simulating $e^{-i\hat{H}t}$ time propagator, that are upper bounded by $\mathcal{O}(\text{Poly}_{\hat{U}}(L, \epsilon^{-1}, t))$, for a time-independent Hamiltonian $\hat{H}$ and up to a simulation time $t$ [6]. Here $\|\hat{H}\|_{\max}$ is the largest absolute element of $\hat{H}$ and $\lambda = \sum_l p_l$ is the sum of all Pauli coefficients of the Pauli-form of Hamiltonian $\hat{H}$.

For the QAS, the quantum computer is used only for estimating the real and imaginary components of the overlap $\boldsymbol{F}$, Hamiltonian $\boldsymbol{H}$, and observable $\boldsymbol{O}$ matrices, as shown in the main text section II A. We choose $n$ time-evolved states $|\psi_j\rangle = e^{-i\hat{H}s_j}|\psi_0\rangle$ as basis, where $j=\{0, 1 \ldots, n-1\}$ and parameter times $0=s_0<s_1<\ldots<s_{n-1}\leq T$ are not longer than the total simulation time. As a result, the aforementioned matrix elements consist of the quantities $F_{jk}=\langle\psi_0|e^{i\hat{H}\Delta s_{jk}}|\psi_0\rangle$ and $P_{jkl}=\langle\psi_0|\hat{P}_l e^{i\hat{H}\Delta s_{jk}}|\psi_0\rangle$ where $k=\{0, 1, \ldots, n-1\}$, $\Delta s_{jk}=s_j-s_k$ are the parameter time differences and $l=\{0, 1, \ldots, 2L\}$. In total, there are $n^2+2Ln^2=\mathcal{O}(Ln^2)$ such quantities combined. We consider using the modified Hadamard test from Appendix B to estimate them. It requires a controlled-$e^{-i\hat{H}\Delta t}$ time propagator that has a longer quantum runtime complexity than a standard $e^{-i\hat{H}\Delta t}$ time propagator by at most a constant factor of $\gamma$. Thus, the overall quantum runtime complexity of the QAS is

$$2 \cdot 2 \cdot L \cdot \sum_{j,k=0}^{n-1}\left[\frac{\Delta s_{jk}}{\Delta t} \cdot \gamma\mathcal{O}(\text{Poly}_{\hat{U}}(L, \epsilon^{-1}, \Delta t))\right] < 4\gamma n^2 \cdot \frac{T}{\Delta t} \cdot L \cdot \mathcal{O}(\text{Poly}_{\hat{U}}(L, \epsilon^{-1}, \Delta t)) \tag{C2}$$

where we used the inequality $\sum_{j,k=0}^{n-1}\Delta s_{jk}<Tn^2$. Comparing the runtime complexities Eqs. (C1) and (C2), we obtain the following condition for the QAS to be more resource-efficient,

$$\frac{T}{\Delta t} \gtrsim 16\gamma n^2 - 1. \tag{C3}$$

By the observation that a 3-qubit Toffoli gate can be decomposed into 6 CNOT gates, we made a reasonable assumption that $\gamma \approx 6$. Thus,

$$\frac{T}{\Delta t} \gtrsim 100n^2 \gtrsim 16 \cdot 6 \cdot n^2 - 1. \tag{C4}$$

**Appendix D: Sampling Basis State Overlaps, Hamiltonian and Observable Elements.**

Here we consider using the standard Hadamard Test to estimate the real and imaginary components of the overlap $F$, Hamiltonian $H$ and observable $O$ matrices using an ancilla qubit, for ease of presentation. The aforementioned matrix elements consist of the following quantities: $F_{jk} = \langle \psi_0 | e^{i\hat{H}\Delta s_{jk}} | \psi_0 \rangle$ and $P_{jkl} = \langle \psi_0 | \hat{P}_l e^{i\hat{H}\Delta s_{jk}} | \psi_0 \rangle$, where $j=\{0,1\ldots,n-1\}$, $k=\{0,1,\ldots,n-1\}$, $l=\{0,1,\ldots,2L\}$, $n$ is the number of basis states, $\Delta s_{jk}=s_j-s_k$ are the time differences, $L$ is the maximum number of Pauli elements in the Hamiltonian $\hat{H}$ or the observable $\hat{O}$ operators, $0=s_0<s_1<\ldots<s_{n-1}\leq T$ are the time parameters. Both $F_{jk}$ and $P_{jkl}$ can be estimated using the Hadamard Test shown in Fig. 5.

The real and imaginary component of the quantities can be estimated by the Pauli-Z expectation value of the ancilla qubit $\langle Z \rangle$ for $b=0$ and $b=1$ respectively. Assuming noiseless quantum circuits and measurements, the $N_s$-shots measurement statistics of the ancilla qubit can be treated as a normal distribution with its mean equal to the Pauli-Z expectation value of the ancilla qubit $\langle Z \rangle$ and its variance $\mathrm{Var}(\hat{Z}) = \sqrt{\frac{1-\langle\hat{Z}\rangle^2}{N_s}}$. Thus, a sample set of the matrices are obtained by sampling the corresponding normal distributions once for each real and an another for each imaginary components of the corresponding quantities respectively.

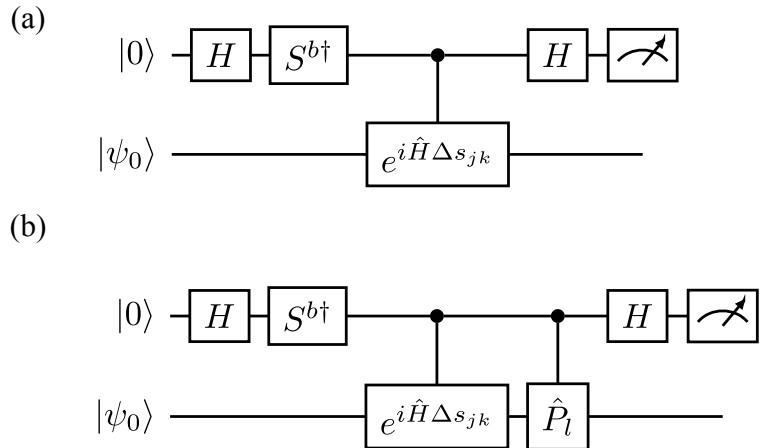

Figure 5. Hadamard test for estimating the real ($b=0$) and imaginary components ($b=1$) of (a) $F_{jk}$ and (b) $P_{jkl}$ quantities.

## Appendix E: Energy Dynamics of Helium Atom and Hydrogen Molecule

We plot the energy dynamics of the He atom and $H_2$ molecule, initialized in an equal superposition of its ground and highest excited state, in Figs. 6 and 7, respectively. The solid colored line and shaded regions represent the mean and uncertainty, respectively, of 100 independent QAS simulation samples, each with $10^4$ simulated shots. The true time-evolution is denoted by the black dashed line. The top left plots of both figures show the total energy which represents the expectation value of the electronic Hamiltonian of the He atom and $H_2$ molecule correspondingly. The top right plots show the Coulomb energy which represents the sum of all interaction energy between two electrons due to Coulomb repulsion. The bottom left plots show the kinetic energy, that is the sum of all one-electron kinetic energies. The bottom right plots show the potential energy which represents the sum of all one-electron potential energy of electrons due to nuclear attraction. For both systems, we observe that the total energy is conserved throughout the entire simulation time, with a fractional energy uncertainty of about 0.8% and 4% for He atom and $H_2$ molecule case respectively. For the Coulomb energy, the fractional energy uncertainty of about 6% and 13% for He atom and $H_2$ molecule case, respectively. For the kinetic energy, the fractional energy uncertainty of about 3% and 5% for He atom and $H_2$ molecule case, respectively. For the potential energy, the fractional energy uncertainty of about 0.5% and 2% for He atom and $H_2$ molecule case, respectively. Despite the seemingly-good accuracy and reasonable precision, some energy oscillations cannot be precisely determined, especially the potential energy plots for both cases. This is due to the energy magnitude being larger than the energy oscillating amplitude. A larger number of shots is required to improve the precision of measuring energy oscillating amplitude.

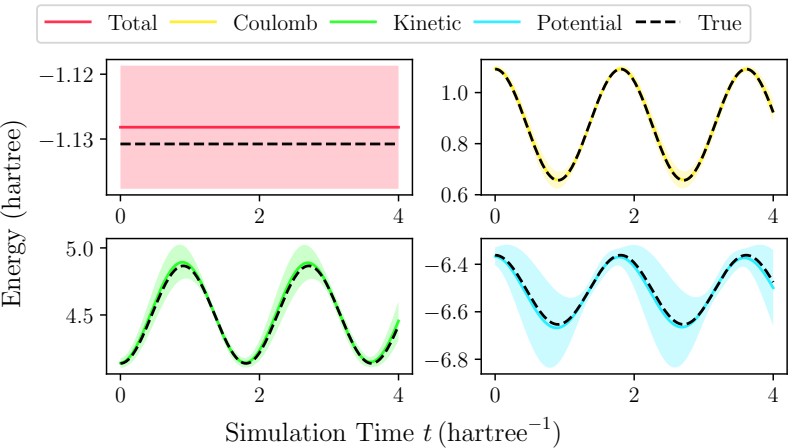

Figure 6. Energy dynamics for Helium Atom, using the 6-31G basis set, initialized in an equal superposition of the ground and highest excited state of eigenenergies -2.87 and 0.609 Hartrees, respectively.

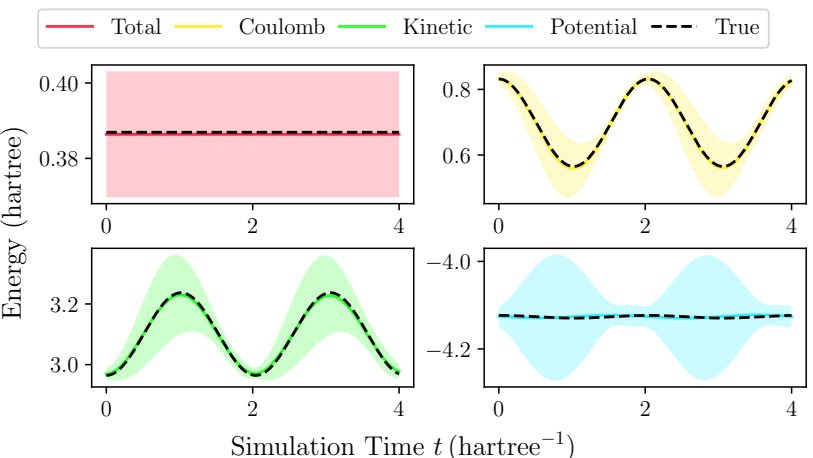

Figure 7. Energy dynamics for Hydrogen molecule, using the 6-31G basis set, at equilibrium distance of 1.4 Bohr, initialized in an equal superposition of the ground and highest excited state of eigenenergies -1.15 and 1.93 Hartrees, respectively.

## Appendix F: Variance of Observed Quantities against Number of Shots

We plot the variance of the orbital population and energy against a range of shots between $10^3$ to $10^{10}$ per real and per imaginary evaluation of basis state overlap and Hamiltonian element, for the He atom in Figs. 8 and 9, respectively, and for the $H_2$ molecule in Figs. 10 and 11. All plots show that the variance is directly proportional to the number of shots, as expected for shot noise.

### 1. Helium Atom Case

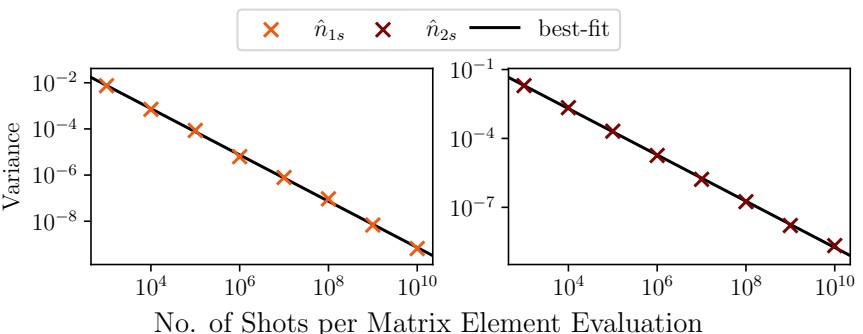

Figure 8. Atomic orbital population variance for 100 independent QAS samples runs for Helium Atom, using the 6-31G basis set.

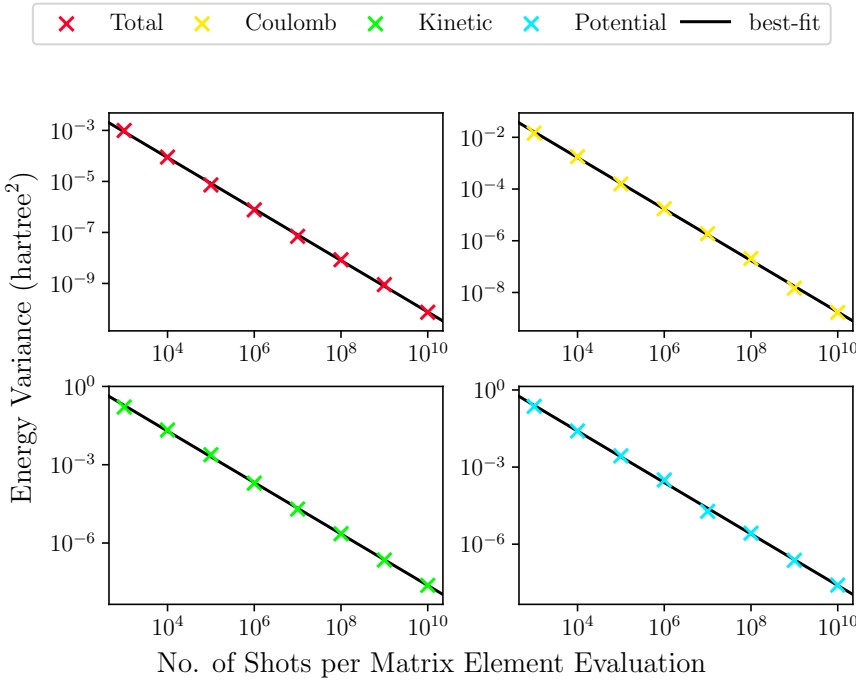

Figure 9. Energy variance for 100 independent QAS samples runs for Helium Atom, using the 6-31G basis set.

## 2. Hydrogen Molecule Case

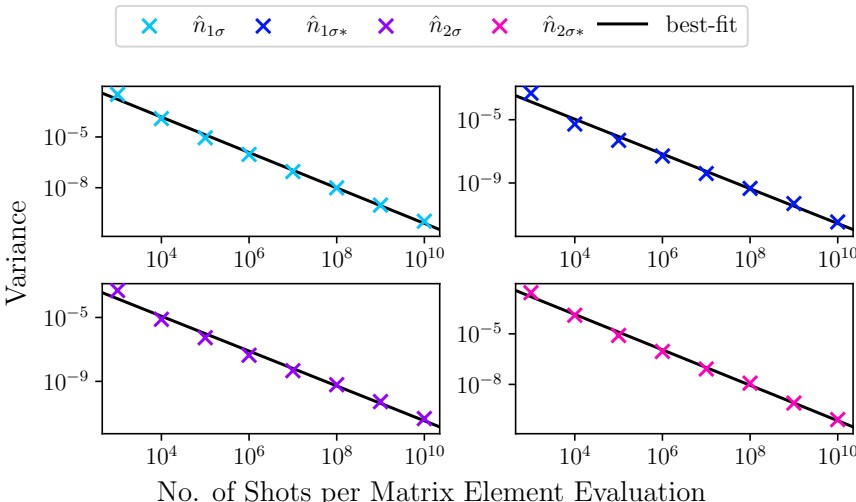

Figure 10. Molecular orbital population variance for 100 independent QAS samples runs for Hydrogen molecule, using the 6-31G basis set.

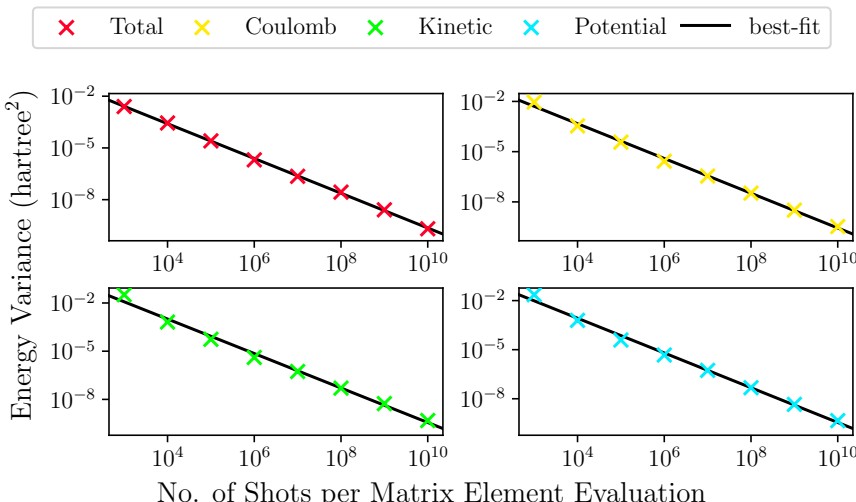

Figure 11. Energy variance for 100 independent QAS samples runs for Hydrogen molecule, using the 6-31G basis set.

## Appendix G: First Order Trotterized Time-Evolved Basis States

We applied first order Trotterization on the basis $|\psi_1\rangle = e^{-i\hat{H}/2}|\psi_0\rangle$ and repeated the QAS simulation. We plotted the state infidelity of the Helium atom problem for a simulation time of $t=4$ Hartree$^{-1}$ against a range of Trotter steps up to $10^4$ steps and a range of shots between $10^4$ and $10^{10}$. The results show that state infidelity improves at larger Trotter steps, but quickly plateaus beyond a finite number of steps as the shot noise becomes the limiting factor.

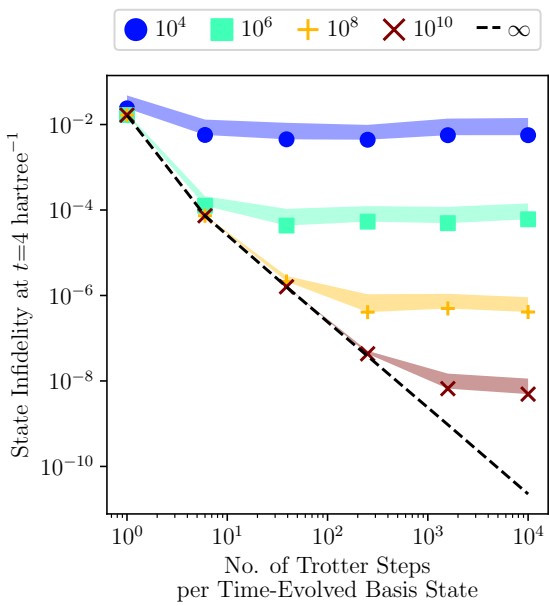

Figure 12. State Infidelity after a simulation time of $t=4$ Hartree$^{-1}$ as a function of the number of first-order Trotter steps implemented per time-evolved basis state, ranging from 1 to $10^4$ steps, at various number of shots ranging from $10^4$ to $10^{10}$ shots per real and per imaginary evaluation of basis state overlap and Hamiltonian element.

## Appendix H: Linear Dependence of the Time-Evolved Basis

The time-evolved basis is linearly independent if and only if the basis state overlap $\boldsymbol{F}$ is invertible. As basis overlap is defined using the $L^2$-inner product, $\boldsymbol{F}$ is also semi-positive definite, that is all of the eigenvalues of $\boldsymbol{F}$ is either zero or positive real-valued. Using the above properties of the basis state overlap, we shall show that for $n=2$ basis set size, the basis linear independence condition for parameter time $s_1$ can be derived. Consider having $n=2$ time-evolved basis, then basis state overlap is

$$\boldsymbol{F} = \begin{pmatrix} \langle\psi_0|\psi_0\rangle & \langle\psi_0|\psi_1\rangle \\ \langle\psi_1|\psi_0\rangle & \langle\psi_1|\psi_1\rangle \end{pmatrix}. \tag{H1}$$

As the basis states are normalized, that is $\langle\psi_0|\psi_0\rangle = \langle\psi_1|\psi_1\rangle = 1$ and the $\boldsymbol{F}$ is a 2-by-2 matrix, we can impose the constraint that $\det(\boldsymbol{F})>0$, to ensure that $\boldsymbol{F}$ is both invertible and semi-positive definite, that is

$$1 - |\langle\psi_1|\psi_0\rangle|^2 > 0. \tag{H2}$$

Suppose the initial state $|\psi_0\rangle$ is a superposition of $n=2$ eigenstates corresponding eigenvalues $e_0$ and $e_1$, that is $|\psi_0\rangle = \beta_0|e_0\rangle + \beta_1|e_1\rangle$, where $|\beta_0|^2 + |\beta_1|^2 = 1$. Also, let the other basis be $|\psi_1\rangle = e^{-iHs_1}|\psi_0\rangle$, then the inequality above in Eq. (H2) can be evaluated to

$$|\beta_0|^4 + |\beta_1|^4 + 2|\beta_0|^2|\beta_1|^2\cos[s_1(e_1 - e_0)] < 1, \tag{H3}$$

which simplifies to

$$s_1 \neq \frac{2k\pi}{e_1 - e_0}, \quad k \in \mathbb{Z}. \tag{H4}$$