# Peer review of "Resource-Efficient Hybrid Quantum-Classical Simulation Algorithm"

_SciPost Physics_

## Round 1 · Referee Report · Anonymous (Referee 1) · 2024-12-11

Strengths

No real strength. The article develops ideas that are already mostly contained in previously published work.

Weaknesses

I have the following poins:
-The article lacks of novelty compared to already published work.
-Lots of discussion in the text are rather standard and useless
-Examples of applications are too simple to really test anything.

Report

The article aims to develop efficient methods for predicting non-equilibrium evolutions in quantum computers. Specifically, the previously proposed QAS is extended using states obtained by the Hamiltonian propagation from an initial state. The method is explained and applied to two examples.

While I do not doubt that the method can be useful, I feel that the ratio between the new aspects is too weak compared to published work. More precisely:
-Using the true Hamiltonian propagated states to project the problems on a reduced subspace has been widely explored in the quantum Krylov method; see for instance
-N. H. Stair, R. Huang, and F. A. Evangelista, A multireference
quantum Krylov algorithm for strongly correlated electrons, J. Chem. Theory Comput. 16, 2236 (2020).
-C. L. Cortes and S. K. Gray, Quantum Krylov subspace
algorithms for ground and excited state energy estimation, Phys. Rev. A 105, 022417 (2022)
( see also arXiv:1909.08925 and the vast literature that cites these articles)

In these articles, similar states are used, and subspace diagonalization is made. Although these articles are focused on spectra. Getting spectra in a reduced space is equivalent to getting the time evolution in this reduced subspace. This could also be understood simply because solving the time-dependent Eq. (4) could be made by first diagonalizing the overlap + Hamiltonian in the subspace. Once the initial state is decomposed onto the approximate eigenstates, getting the evolution is straightforward.

Besides this main comment, I have also some more remarks:
1) The illustrations given in the article seem pretty simple. Indeed, for instance, the Helium atom, if I understood correctly, is treated on four qubits. Then, the number of Slaters determinants to treat the problem is 6. If, in addition, the Hamiltonian has some specific block structure due to symmetries and the initial state belongs to one of the blocks, the number of true eigenstates on which this state might be highly reduced. The oscillatory pattern of Fig. 2 indicates that 2 states might be enough, and not surprisingly, n=2 seems a very good approximation of the approach. If this is the case, using or not time-propagated states does not make any difference compared to using any other states. Related to this aspect, n is compared to 2^n in many places, which is irrelevant since 2^n corresponds to the full Fock space.
2) Appendix A and C are pretty standard aspects of quantum computing and are unnecessary to recall. Also, I am a bit confused about appendix F. Is the scaling found not simply the standard scaling with the number of shots that scales as the square of the number of shots? If this is the case, the appendix is not meaningful.

In summary, I estimate that the article's content is insufficient, and not new enough to justify publication.

Requested changes

I do not recommend the article in any form.

Recommendation

Reject

---

## Editorial Decision

in_refereeing